# Mass enhances speed but diminishes turn capacity in terrestrial pursuit predators

Rory P Wilson[1]*, Iwan W Griffiths[2], Michael GL Mills[3,4], Chris Carbone[5], John W Wilson[6], David M Scantlebury[7]

[1]Swansea Lab for Animal Movement, Department of Biosciences, College of Science, Swansea University, Swansea, Wales; [2]College of Engineering, Swansea University, Swansea, Wales; [3]The Lewis Foundation, Johannesburg, South Africa; [4]Wildlife Conservation Research Unit, Department of Zoology, University of Oxford, Oxford, United Kingdom; [5]Institute of Zoology, Zoological Society of London, London, United Kingdom; [6]Department of Zoology and Entomology, University of Pretoria, Pretoria, South Africa; [7]School of Biological Sciences, Institute for Global Food Security, Queen's University Belfast, Belfast, United Kingdom

**Abstract** The dynamics of predator-prey pursuit appears complex, making the development of a framework explaining predator and prey strategies problematic. We develop a model for terrestrial, cursorial predators to examine how animal mass modulates predator and prey trajectories and affects best strategies for both parties. We incorporated the maximum speed-mass relationship with an explanation of why larger animals should have greater turn radii; the forces needed to turn scale linearly with mass whereas the maximum forces an animal can exert scale to a 2/3 power law. This clarifies why in a meta-analysis, we found a preponderance of predator/prey mass ratios that minimized the turn radii of predators compared to their prey. It also explained why acceleration data from wild cheetahs pursuing different prey showed different cornering behaviour with prey type. The outcome of predator prey pursuits thus depends critically on mass effects and the ability of animals to time turns precisely.

*For correspondence: r.p.wilson@swansea.ac.uk

**Competing interests:** The authors declare that no competing interests exist.

**Reviewing editor**: Iain D Couzin, Princeton University, United States

## Introduction

Animals that feed on mobile prey have evolved a repertoire of anatomical features and behavioural strategies to maximize capture success (*Lima, 2002*), matched by prey in an evolutionary arms race in their attempts to avoid capture (*Randall et al., 1995*; *Walker et al., 2005*; *Cortez, 2011*). Numerous authors have examined strategies for both predators and prey in such interactions (*Weihs and Webb, 1984*; *Domenici et al., 2011*), with a particular wealth of literature on animals operating in air (*Warrick, 1998*; *Hedenström and Rosén, 2001*) and water (*Domenici, 2001*) with considerations ranging from turn capacity (*Fish, 1999*), through random turning behaviour by prey (*Jones et al., 2011*; *Combes et al., 2012*) to speed accuracy tradeoffs during decision making (*Chittka et al., 2009*). Such considerations appear less well developed in terrestrial, cursorial interactions with, in particular, no discussion of how varying mass between parties might affect strategy, although this has been examined in the aquatic literature (*Domenici, 2001*).

*Elliot et al., (1977)* provided a conceptual model for prey acquisition by terrestrial carnivores describing four major elements; the search, the stalk, the attack, and the subdue. Of these, the attack is the most power-demanding (*Williams et al., 2014*), typically involving complex high speed manoeuvres (*Van Damme and Van Dooren, 1999*; *Kane and Zamani, 2014*), underpinned by apparently complicated behavioural options for both predators and prey (*Estes and Goddard, 1967*;

**eLife digest** A pursuit between a predator and its prey involves complex strategies. Prey often make sudden sharp turns when running to evade a predator. Any predator that cannot turn quickly enough will have to run further to catch up with the prey again, thus potentially allowing the prey to pull away from the predator. The timing of these turns is crucial; if the prey turns when the predator is too far away, the predator can cut the corner off the turn and catch up with the prey more easily.

The speed at which animals can turn depends on the forces involved in cornering, and larger animals need to produce greater forces for any given turn. However, larger animals can apply relatively less force than smaller animals for turns and so cannot turn as rapidly. The effect of the relationship between mass and turning ability on the strategies used during land-based pursuits had not been investigated.

Wilson et al. have now created a mathematical model that considers how the mass of a predator and its prey influences the course and strategies used in a land-based pursuit. The model is based in part on a mathematical problem called the 'homicidal chauffeur game', where a car driver attempts to run over a pedestrian. Wilson et al.'s model predicts that chases between large predators and smaller prey should feature frequent sharp turns, as the prey try to exploit their superior turning ability. However, when the predators and prey are of similar size, the prey gain little or no advantage from executing high-speed turns. Indeed, as turning slows the prey down, turning may often be disadvantageous, and so fewer turns should be seen during a pursuit.

The predictions of the model were compared with the pursuit strategies of wild cheetahs, which were studied using collars equipped with tags to measure acceleration as the predators chased prey of different sizes—from hares to large antelopes called gemsboks. The tracking data confirmed the predictions of the model; thereby revealing that body mass and the ability of animals to choose when best to turn strongly determine the outcome of predator-prey pursuits.

*Wilson et al., 2013b*). In fact, operating in a planar terrestrial environment, options for both parties are restricted, being based primarily on the choice of speed (*Elliot et al., 1977*) and/or trajectory (*Howland, 1974*), with actual performance in these being determined by physical constraints that determine maximum attainable speed and limits on turn radius as a function of velocity.

Maximum speed in terrestrial running animals tends to increase to an asymptote with body mass (*Garland, 1983*) and this process is driven by increases in leg length with body size, which facilitates higher speeds (*Garland, 1983*; *Bejan and Marden, 2006*), being ultimately limited by the scaling of body mass relative to leg strength (*Alexander, 2002b*).

Turn radius as a function of velocity in terrestrial animals is modulated by a number of critical elements, notably the extent to which leg strength can support the forces required to generate the centripetal acceleration for the turn (*Greene and McMahon, 1979*; *Greene, 1985*; *Alexander, 2002b*; *Tan and Wilson, 2011*). In addition, important elements influencing turning mechanics and maximum turn speed are; the interplay of limb force limits (*Chang and Kram, 2007*) and friction limits (*Usherwood and Wilson, 2005*, *2006*), and morphological factors (*Walter, 2003*; *Jindrich et al., 2007*; *Jindrich and Qiao, 2009*).

Across these studies on terrestrial animals, however, the most consistent factor limiting speed during turns is force limits, with the total force demands relating to the combined effects of supporting the body mass and providing the necessary acceleration. This means that there must be a broad scaling trend in turning performance with body mass. To our knowledge though, although such scaling trends have been considered for animals operating in water and air (*Domenici, 2001*; *Hedenström and Rosén, 2001*), this has not been explicitly examined in the literature for terrestrial animals. This trend is due to the mismatch between the linear relationship of increased force demands with body size (proportional to $Mass^{1.0}$), and the non-linear scaling of leg strength (roughly proportional to $Mass^{0.66}$ (*Schmidt-Nielsen, 1984*) (but see *Biewener, 1989*), leading to a relative decrease in strength, and therefore turn capacity, with increasing size. The seminal work on turn performance (*Greene and McMahon, 1979*; *Greene, 1985*; *Usherwood and Wilson, 2005*, *2006*; *Tan and Wilson, 2011*) has, to date, essentially concentrated on individual species, where variation with mass is of little consequence except in an absolute sense. However, where species interact, we

would expect this mismatch to have profound implications, and perhaps nowhere more than in predator-prey interactions during pursuit.

This work investigates the implications of mass in modulating options for terrestrial mammalian predators during the attack phase of attempts at prey capture. We limit ourselves to terrestrial predation because mass effects have already been examined for organisms in fluid media (*Domenici, 2001*; *Van den Hout et al., 2010*) and because dealing with animals that operate in a 2-dimensional surface obviates many of the complexities associated with the energetics of changing height in aerial animals (*Weihs and Webb, 1984*). In order to tease out expected trends, we use a simple model that isolates only the features of mass relevant to force demands, and mass-linked capacity for force production. Thus, we assume, among other things, that both predators and prey are geometrically similar (which ignores compensating mechanisms such as upright posture and more robust limbs of some large animals as well as variation in traction) and that the chase environment is flat, open and homogeneous (*Howland, 1974*; *Wilson et al., 2011*). We propose that the attack phase of pursuit predators is essentially what has been treated within theory encompassing the 'homicidal chauffeur' game (*Marec and Van Nhan, 1977*), where a car driver attempts to hit a pedestrian in a defined open space. In this, we divide trajectories of both parties into straight line and turn sections and, by adopting a game theory-based approach (*Dugatkin and Reeve, 1998*; *Lima, 2002*), we distil out simple objectives for both parties; predators should attempt to minimize their distance to the prey, while prey attempt to maximize this distance (*Weihs and Webb, 1984*). To date, the most notable attempt to define the predator-prey chase scenario is the work by *Howland (1974)*, who defined many of the rules and consequences for a single turn gambit and we use this work as a starting point. However, a large body of theory also exists (notably *Wei et al., 2009* and references therein), particularly that dealing with the best strategy for missiles to engage with their targets to maximize strike probability (*Shneydor, 1998*; *Siouris, 2004*) and this is also considered within our terrestrial pursuit scenario.

Specifically, our work has four elements. Firstly, for predator and prey mass equivalences, we examine how the initiation of a turn by a prey being pursued by a predator affects the change in predator-prey distance according to speed and timing, before extending the single turn scenario up to multiple turns. Secondly, we consider how differential masses between predators and prey affect the outcome of single turn manoeuvres. Third, we then use acceleration data acquired from tags deployed on free-living cheetahs pursuing prey of varying masses (*Hayward et al., 2006*) to consider whether our model predictions are broadly manifest in the wild. Finally, we compile data on the masses of mammalian predators and their prey to examine whether our model explains general patterns in prey size selection (*Carbone et al., 2007*).

Our approach reveals that the dynamics of movements by predators and prey of varying body masses can be treated within a single framework where the classifications and likely outcomes of pursuits, as well as the relative sizes of predators and prey, seem largely dependent on simple physical rules.

## Results

### The predator-prey pursuit model

Our model for deriving the characteristics of a defined turn as a function of speed in equivalent-sized predators and prey predicted that single turns initiated by the prey lead to one of primarily two phenomena: Either the predator cuts the corner, reducing the predator-prey distance (*Figure 1A*) benefitting the predator, or it overshoots the corner, increasing the predator-prey distance (*Figure 1B*) benefitting the prey. A third scenario might be where the predator follows the prey trajectory precisely, in which case there is no change in benefit to either party although in this circumstance, the predator, with its higher speed, will eventually converge on the prey (*Figure 1C*). Although a single turn may not lead to capture or escape of the prey, multiple turns with consistent undershooting by the predator can do so (*Figure 1*). For any given turn, the change in predator-prey distance over time was predicted to be critically dependent on; (i) the distance between the parties at the moment of the turn (*Figure 2A*), with shorter distances (for distances >0) at the moment of the turn leading to greater overshoot by the predator, (ii) the difference in speed between parties (*Figure 2B*), with greater speed differences leading to greater overshoot by the predator, and (iii) the reaction time of the predator (*Figure 2C*), with slower reaction times leading to greater overshoot.

Where predators and prey have different masses, the model predicted mass-dependent distances travelled in a given turn, with larger animals having to run farther during cornering (*Figure 3*). Thus,

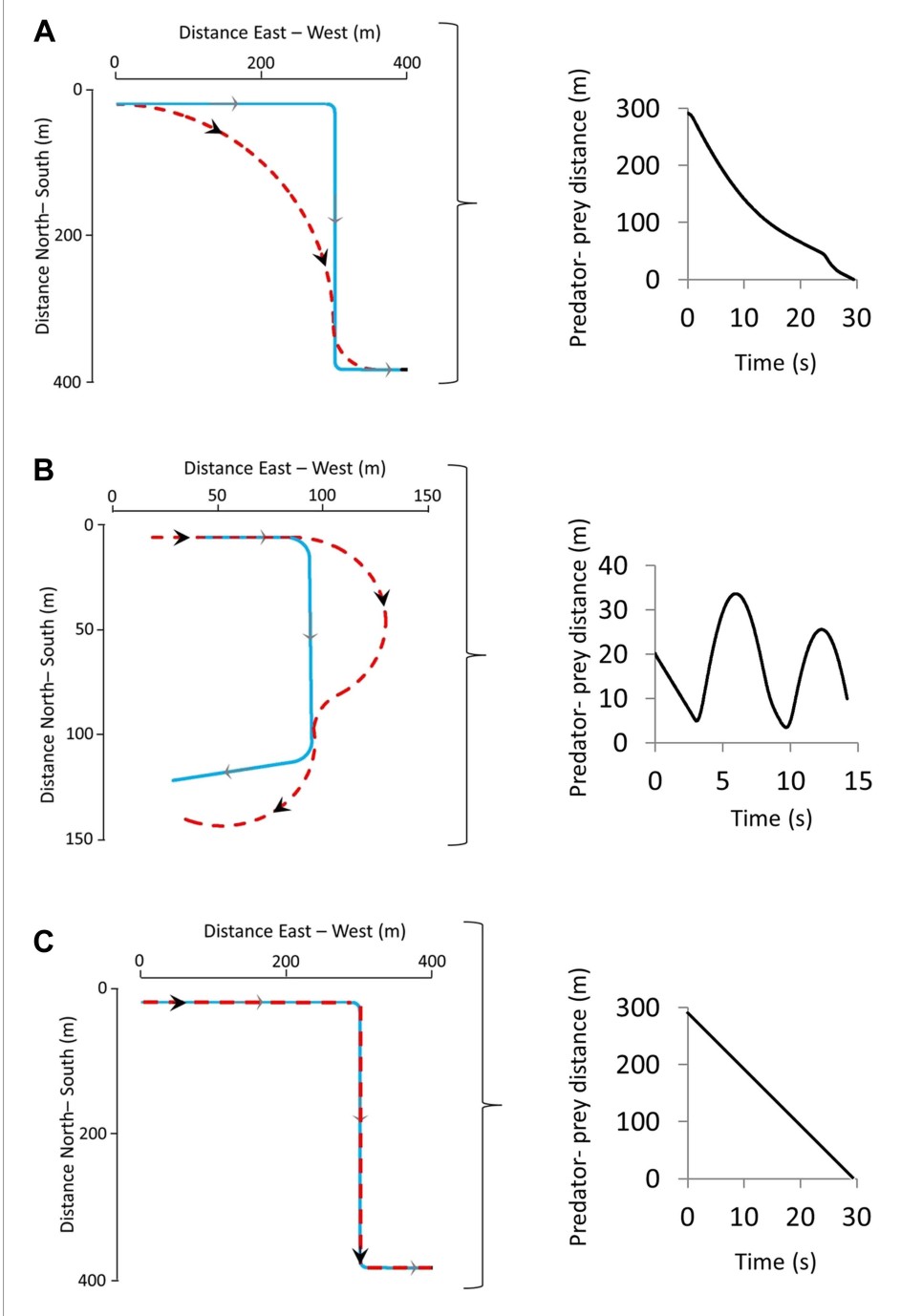

**Figure 1**. Two sequential prey turns during predator-prey pursuits showing the trajectories (left-hand figures) adopted by both prey (blue lines) and predator (red lines) during instances of turns that are (**A**) too early or (**B**) correctly timed by the prey, leading to corner-cutting or overshooting, respectively, by the predator. (**C**) shows the scenario where the predator and prey adopt identical trajectories. The right hand figures show how the distance between the predator and prey varies with time for the shown trajectories.

during a single 90˚ turn, a 250 kg predator (an example of which may be a lion or tiger) is predicted to have to run farther than all considered prey (potential prey ranging between a 3 kg and a 200 kg), whereas a 30 kg predator (e.g., a cheetah or a wolf) is predicted to travel shorter distances during a turn than 3 of the 5 prey masses considered (*Figure 3*). Thus, cornering is predicted to be more

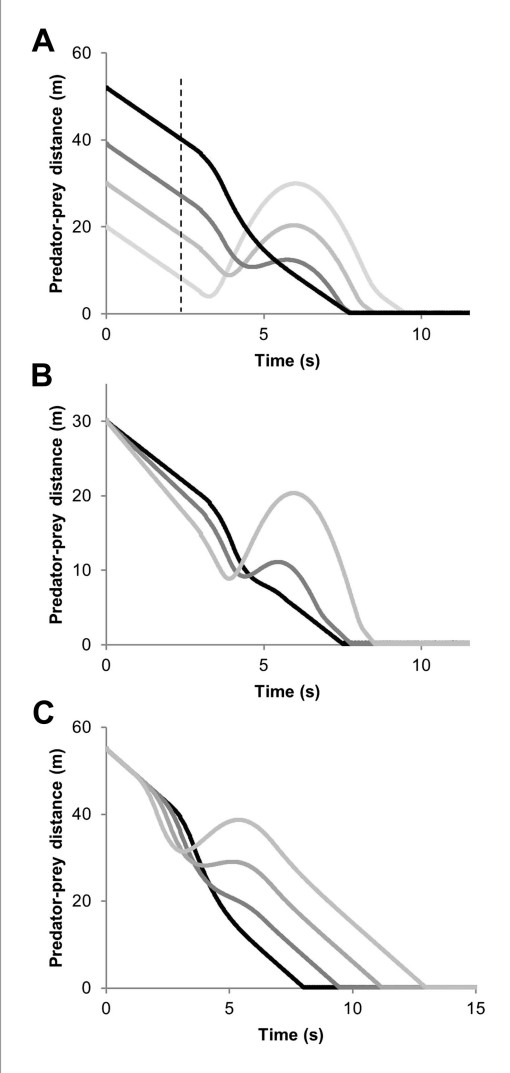

**Figure 2**. Predator-prey distance during pursuit as a function of time during an execution of an unexpected 90° turn by the prey followed by straight-line travel, for; (**A**) different predator-prey distances at the moment of the execution of the turn by the prey (indicated by the dashed vertical line) [reaction time of the predator = 0.3 s, predator and prey speeds 20 and 15 m/s, respectively], (**B**) different predator speeds (black line shows a speed of 18 m/s and increasingly pale lines show speeds of 19 and 20 m/s respectively) [reaction time of predator = 0.3 s ] (**C**) different reaction times by the predator (black line shows a reaction time of 0.5 s and increasingly pale lines show reactions times of 1.0, 1.5 and 2.0 s, respectively) [predator and prey speeds = 20 and 15 m/s, respectively] .

advantageous as an escape manoeuvre as the prey size decreases relative to that of the predator and we would expect to see evidence of that in wild animal data.

This prediction could be examined in the data derived from the fieldwork on free-living cheetahs, where we observed 36 pursuits involving 7 prey species (*Table 1*), of which 33 had corresponding acceleration data for 5 species totalling 899 s.

Identifying cornering behaviour via lateral g-forces, we documented a total of 547 turns within all chases from all animals. In these, there was a significant interaction between prey species and turn number (turns were sequentially numbered within each chase) on the rate of turn: specifically, different prey species had different turn rates as the turn numbers progressed (*Figure 4*) ($\chi^2$ = 11.13, p = 0.03, df = 4) (*Table 1*). We also noted that cheetah turns became shorter as the chase progressed (Spearman's ρ = −0.347, p < 0.0001, although g-values reached during turns did not change significantly over time (ρ = 0.031, p = 0.72). Critically, these dynamics were different in successful and unsuccessful pursuits. Turn duration decreased with increasing turn number within any particular chase ($\chi^2$ = 26.52, df = 1, p < 0.0001) and turn durations were shorter in successful pursuits ($\chi^2$ = 6.24, df = 1, p = 0.013).

Extending our modelling exercise to derive maximum cornering ability as a function of mass and speed for a suite of theoretical mammalian cursorial predators and their prey indicated that, within the animal mass range considered, mass affects maximum speed (by a factor of less than 6) much less than it affects minimum turning radius at maximum speed (which affects it by a factor of up to 260). This is due to the combined effect of generally increasing maximum speeds with mass (see above) and the additional effect of mass on turn radius (*Figure 5*). Importantly though, the surface describing comparative turn abilities showed that predators turn tighter relative to prey in a specific area of the surface defined in terms of predator and prey mass ratios (*Figure 5*). Insertion of the mean mass of 54 species of canids and felids and their (predominantly mammalian) prey from an extensive database (*Carbone et al., 1999*) into this plot, shows that all predators were located in the area where predator turn capa-

bilities were maximized compared to their prey (*Figure 5*). The implication from this is that there is strong selection pressure for turning ability in predators and that (i) predators evolve to take particular sized prey with a mass that results in the minimum turn radius ratio most in favour of the predator, and/or (ii) prey sizes that have minimum turn radii that most closely accord with those of the predators tend to be caught more often than other-sized prey.

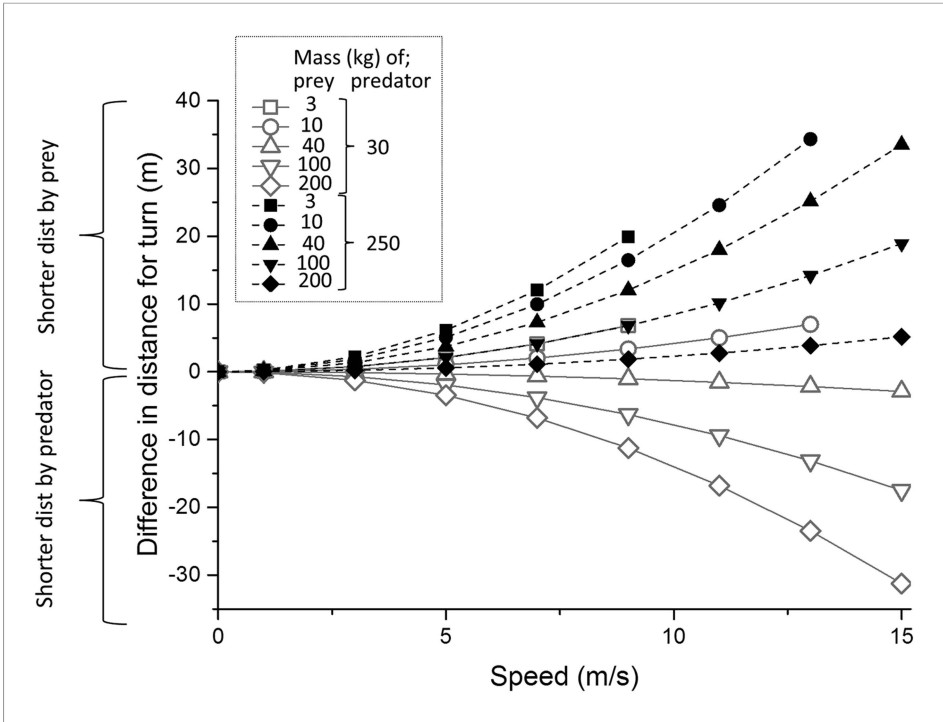

**Figure 3**. Predicted difference in distance travelled during a 90° turn by two different predators, one of mass 30 kg (e.g., a cheetah or wolf—continuous grey lines) and one of mass 250 kg (e.g., a lion or tiger—black dashed lines) compared to that travelled by prey of various masses (indicated by different symbols) as a function of running speed. Prey masses might correspond to for example, 3 kg—a hare, 10 kg—a steenbok, 40 kg—a springbok, 100 kg—a white-tailed deer, 200 kg—a hartebeest. Positive values show a greater distance run by the predator, negative values show greater distance run by the prey. Note that not all speeds reach 15 m/s due to the smallest prey not being predicted to reach this maximum (see text).

## Discussion

### Chase strategies

The outcome of a predator-prey pursuit depends on performance, both in terms of speed and cornering capacity (*Wilson et al., 2013a*, *2013b*), and how these relate to power requirements and

**Table 1**. Summary of the characteristics of cheetah-prey pursuits with prey nominally ranked in order of mass (top smallest to bottom largest)

| | Chase parameters | | | | | | | |
| | No. hunts | Success | Total duration (s) | | No. of turns | | Turn rate (Hz) | |
| Prey | N | % | Mean | SD | Mean | SD | Mean | SD |
|---|---|---|---|---|---|---|---|---|
| Hare | 2 | 100 | 10.8 | – | 5 | | 0.5 | 0.26 |
| Steenbok | 19 | 53 | 28.7 | 11.9 | 5.47 | 2.97 | 0.32 | 0.19 |
| Duiker | 1 | 100 | – | – | – | – | – | – |
| Springbok | 7 | 57 | 28.3 | 7 | 6.17 | 3.06 | 0.32 | 0.17 |
| Ostrich | 1 | 100 | – | – | – | – | – | – |
| Wildebeest | 2 | 50 | 35 | 3.2 | 6.5 | 0.71 | 0.22 | 0.09 |
| Gemsbok | 4 | 75 | 18.6 | 10.6 | 2.33 | 1.53 | 0.15 | 0.05 |

Two species were pursued where no corresponding acceleration data were available.

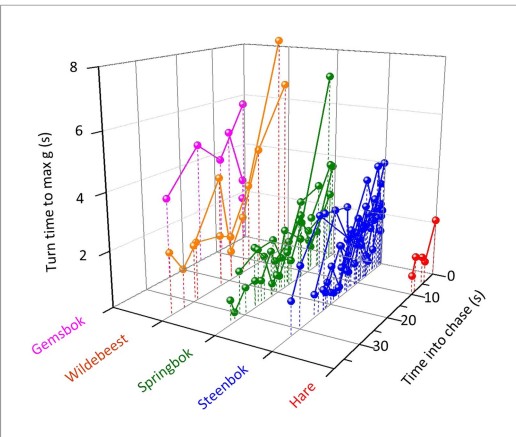

**Figure 4**. The duration of turns (expressed as the length of time between the onset of the turn and the point at which maximum g was reached within the turn) made by cheetahs pursuing different prey with time into the chase. Joined points show chases referring to individual prey. Longer turns will tend to have larger turn radii so while, overall, pursuit of larger prey is characterized by larger turn radii, turn radii diminish as the chase progresses.

timing. Our work echoes that of *Howland (1974)*, which dealt with the same issue using a different approach, emphasising that high speed, aside from being necessary for the predator to gain on the prey, also increases the likelihood that the predator will overshoot the prey turn trajectory (*Figure 1B*) (*Howland, 1974*; *Alexander, 2003*). Conversely, as speed decreases, the difference in distance covered during any turn between predator and prey decreases (*Figures 1–3*), fulfilling the fundamental game rule for the predator. The optimum strategy for a pursuit predator should, therefore, be to attempt to elicit turns by the prey which, if the prey are working close to maximum power, will result in a reduction in their speed (*Shubkina et al., 2012*), because energy is needed for the turn (*Wilson et al., 2013c*). Invoking multiple rapid turns might therefore be a strategy that predators seek to promote. Here, decreasing turn duration over time is expected during successful chases, as we observed (*Figure 4*), coupled with generally decreasing speed, as reported by Wilson et al. (*Wilson et al., 2013b*). However, turning is also critical for survival of prey because maintenance of a straight line trajectory leads to inevitable capture if the predator is faster. This seemingly contradictory situation of whether the predators or prey benefit most from prey turns is clarified by timing. When the predator is far from the prey, the prey should not turn since to do so allows the predator to cut the corner of the prey's trajectory and decrease the distance between itself and the prey more rapidly (*Eilam, 2005*) (*Figure 1*). However, turns by the prey do benefit the prey if the timing is correct because this leads to an overshoot by the predator (*Figure 1*, *Figure 2*). Where such overshoot turns are consistent, they should generally lead to rarefying turn rates with reduced chances of capture (*Wilson et al., 2013b*).

The outcome of extended pursuits is also likely influenced by endurance. Both parties will have power limitations restricting their options on instantaneous performance, as well as ultimately limiting how long the chase can continue before exhaustion. A multiple turn terrestrial chase has energetic costs associated with straight line travel that are a (linear) function of speed (*Taylor and Heglund, 1982*) so, all other things being equal, at such times the predator should be expending energy faster if it is to gain on the prey. However, there are substantial costs to turning above those of straight line travel (*Wilson et al., 2013c*) so the angular extent of the turn and the time spent turning will both affect the rate of energy expenditure (*Wilson et al., 2013c*). Thus, where the predator cuts the corner compared to the trajectory taken by the prey (*Figure 1*), the reduced turn costs should act to reduce its overall rate of energy expenditure, making the predator energy expenditure closer to that of the prey. This will tend to lead to similarity in power use between parties resulting in similar giving-up times, assuming both parties can allocate similar amounts of energy to a pursuit (*Figure 6*). However, where a predator overshoots the cornering trajectory of the prey (*Figure 1*), it has to contend with the increased energetic demands of travelling farther, and with a greater turn angle, than the prey (*Figure 2*). Thus, where overshooting occurs consistently, it will tend to make cumulative energy expenditure between the two parties more disparate resulting in the predator reaching endurance limits earlier than the prey (*Figure 6*). In reality, multiple turn pursuits, such as we observed in our cheetah-prey interactions, will consist of a both corner-cutting and overshooting by the predators with the proportion of either perhaps biasing the likelihood of prey capture or the chase being abandoned.

## The role of mass in chase strategies

Our data and theoretical considerations based on the literature highlight the extent to which mass drives physical abilities in predator-prey pursuits. The fact that maximum speed generally increases

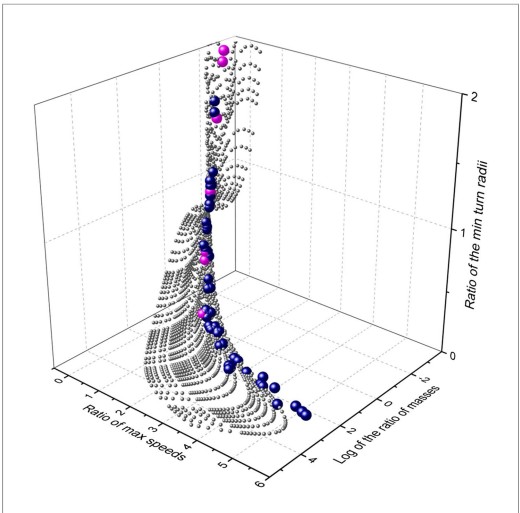

**Figure 5**. Relationship between mass and performance for predators and prey derived from the model. Performance is expressed as maximum speed ratio (Vmax pred/Vmax prey; higher values indicate relatively faster predators) and minimum turn radius (Tmin prey/Tmin pred; higher values indicate relatively tighter turning predators). The grey spheres indicate the range of possible options computed systematically, the blue spheres represent single predator species computed against their mean prey masses, while the cyan spheres indicate the specific case of the cheetah, taking variously sized prey items across the size range.

with mass (*Garland, 1983*) has been variously linked to factors such as absolute and relative leg length, stride length and stride frequency (cf. *Schmidt-Nielsen, 1984*; *Biewener, 1989*, *2003*; *Alexander, 2002a*). This is complicated by the degree of geometric similarity between animals and models for elastic similarity (cf. *Garland, 1983* and refs therein), all factors which might also help explain the modest reverse trend in the maximum speed vs mass relationship for animals exceeding about 70 kg (*Garland, 1983*). That greater mass leads to greater turn radii, because the forces that animals can exert for a turn relate to a mass exponent of less than one, whereas the forces required for a turn scale to a mass exponent of one has not, to our knowledge, been previously discussed. Both these mass-dependent attributes lead to a tendency for larger animals to be faster, but less able to turn than smaller animals, which presumably has profound consequences for strategies adopted by predators and prey during pursuits.

Mass-linked performance explains why, for example, smaller cheetah prey confer a greater size-derived speed advantage to the predator, which should lead to a more rapid closure of the cheetah-prey distance during straight line sections of the pursuit and result in a predicted, and observed, higher turning frequency in cheetah pursuing smaller prey (*Figure 4*, *Table 1*). Similarly, our mass-dependent model of performance points to how predator size in relation to that of the prey results in differential distances travelled by both parties during turns. So the distance run during any given turn by a predator with similar mass to its prey should not be markedly greater than its prey, while a larger predator chasing the same prey must contend with covering a substantial increase in the distance travelled (*Figure 3*). However, the scaling of turning radius with speed is complex because either party may elect to travel more slowly to produce a tighter turning radius, which also reduces the chances of mistakes (*Chittka et al., 2009*), although the predator must always presumably travel faster than the prey. Certainly, workers have suggested (*Wilson et al., 2013a*), and found (*Wilson et al., 2013b*), that cheetahs tend to reduce speed in the final stages of their chases (cf. *Shubkina et al., 2012*).

Thus, varying masses between predators and prey changes the nature of pursuits substantially. Pursuit predators up to body masses of about 70 kg would seem to need to be larger than their prey in order to be able to catch them because, up to this size, larger animals can run faster (*Garland, 1983*; *Bejan and Marden, 2006*). But larger-bodied predators must modulate speed carefully to compensate for their turn radius disadvantages because, although they can outrun, they are less likely to out-manoeuvre their prey (*Figures 3, 4*). Therefore, chases are expected to be defined by more frequent turns with increasing mass differential between parties, which is what we observed in our cheetah data (*Table 1*).

Where predators pursue prey larger than themselves, however, and assuming that the predator can travel faster than their prey, there is no advantage to be gained by the prey executing the sudden turns characteristic of small prey (cf. *Figure 4*). This explains the straight-line trajectories of for example, gemsbok *Oryx gazelle* being pursued by spotted hyaenas *Crocuta crocuta* (*Mills, 1990*). The primary deterrent to smaller predators hunting larger animals may come from the danger of injury (*Mills, 1990*), or the predator not having the strength to overcome its prey, something that can be mitigated to an extent by co-operative hunting or following the prey to exhaustion (*Estes and Goddard, 1967*).

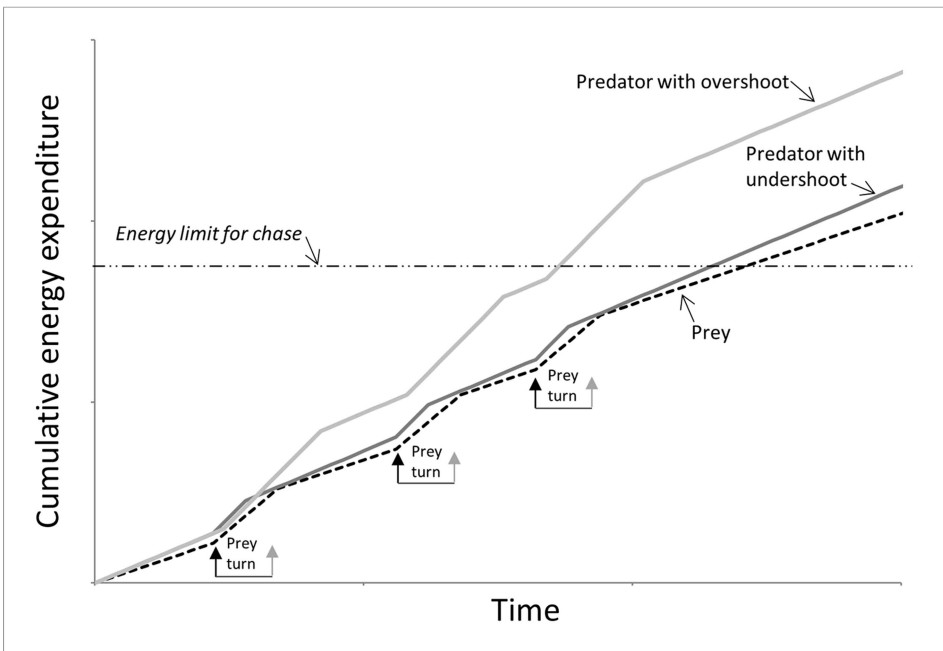

**Figure 6**. Schematic figure to show how power use is expected to vary during the course of a predator-prey chase consisting of four straight-line trajectories interspaced with three turns (turns made by the prey are shown between arrows) by both parties. The prey (dashed line) has lower energy expenditure than the predator during straight-line sections because it is travelling slower. However, because extra power is required for a turn (*Wilson et al., 2013c*), a predator that consistently cuts the corner (dark grey line—'predator with undershoot'—cf. *Figure 1*) spends less time cornering, expending less energy for the corner, and may maintain energy expenditure at levels similar to those of the prey despite travelling faster: Here both parties may reach limits to endurance performance at a similar time. However, a predator that consistently overshoots the corner (light grey line—'predator with overshoot'—cf. *Figure 1*) spends longer turning, expending markedly more energy than the prey at all times, reaching endurance limits earlier. (In this depiction, predators and prey are assumed to have the same geometry, performances and masses).

Our treatise assumes that both predators and prey interact on a homogeneous, flat surface but we expect any variation in the topography and vegetation to modulate the tactics adopted by both predators and prey. *Shepard et al., (2013)* note how substrate type affects the costs of animals moving over it (hence their term 'energy landscape'), which is predicted to affect route choice in a general sense. We expect high power pursuits of the type discussed here to be subject to the same rules, with changes in the energy landscape that differentially affect predators and their prey to be exploited by the relevant party. Of particular note is the work by *Taylor et al., (1972)*, who noted that larger animals incur a proportionately greater increase in metabolic rate for movement up inclines, and this has been shown to affect area use in some species (*Wall et al., 2006*). Correspondingly, we would expect smaller prey to favour selection of uphill gradients during pursuits. Similarly, smaller prey are expected to 'run for cover' (*Domenici et al., 2011*), partly because such cover may represent an impossibly high-cost energy landscape for the predators. In addition, vegetation in patches, such as bushes or trees, may constrain turn radii, precluding small prey from perhaps turning as tightly as they might, to the advantage of the predator while, conversely, such features may allow prey to execute a turn without giving the predator the option of cutting the corner (*Figure 7*).

Within the general context of predator-prey pursuits, some authors have noted that unpredictable (protean) movement by prey can enhance their chances of escape (*Jones et al., 2011*). The theory is that unpredictable movement may catch the predator by surprise (*Humphries and Driver, 1970*), not least because, in our case of cursorial predator-prey interactions, the prey may execute a turn when the predator-prey distance is not yet critical. Clearly, if this occurs in planar environments, choices for changing trajectory can only amount to movement that is either left or right, and should not be executed too early where there are substantial corner-cutting benefits to be gained by the predator.

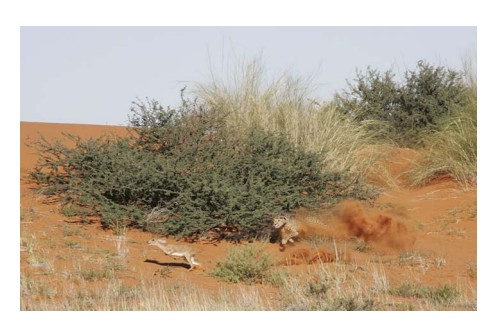

**Figure 7**. Pursuit of a steenbok which has turned as it ran past a bush, constraining the cheetah to follow almost the same trajectory. The use of environmental features such as this makes the timing of the turn less critical since the cheetah cannot cut the corner substantially, even if the prey turns too early.

However, since the timing of corners would appear so critical (see above), we would expect selection pressure for random turns just before the critical phase. Future work would do well to consider this. Finally, predator-prey pursuit options by terrestrial animals will undoubtedly be altered when the condition changes from single to multiple predators. Here, observed manoeuvres are complex and include fanning out of predators (*Kelley, 1973*; *Mills, 1990*) which should allow larger predators to mitigate for the effects of their reduced turning radii with respect to those of their prey. Indeed, understanding the precise advantages of group hunting as a function of group size, predator and prey masses, and the distribution of all parties in space in relation to environmental variability promises to be a major challenge.

## Materials and methods

### Providing a framework for a terrestrial predator-prey pursuit model

There are a number of different strategies recognised within 'pursuit' scenarios (e.g., athletes chasing balls, children's games of 'tag', missiles, and insects involved in territorial disputes) including, primarily 'pure' pursuit, and more 'predictive' strategies such as 'constant bearing' pursuit, and 'constant absolute target direction' (CATD) pursuit (for definitions see *Shneydor, 1998*; *Wei et al., 2009*; *Nahin, 2012*).

Constant bearing is only applicable if the speeds of both parties are constant (*Reddy, 2007*), which is generally inappropriate for predator-prey terrestrial pursuits (*Wilson et al., 2013b*) although its use has been demonstrated in some predatory insects (*Olberg et al., 2000*), fish (*Lanchester and Mark, 1975*) and humans (*McBeath et al., 1995*), while CATD has been shown in some birds, as has 'pure pursuit' (*Kane et al., 2015*). We expect, however, pursuit strategies for flying animals (and some latter-day missiles) to differ from those of terrestrial animals. This is because initial engagement between two aerial parties invariably involves both moving prey and the pursuer having an approach that tends to occur at a tangent to the target (*Hedenström and Rosén, 2001*; *Ghose et al., 2006*), rather than from directly behind. Part of the reason for this tangent relates to both predators and prey operating in 3-d, rather than 2-d, space so that the simple probability of a line of attack being directly along and behind the prey flight path at first encounter is correspondingly reduced. Tangential approach provides various options for pursuers to calculate trajectories and compute intersection points (*Olberg et al., 2000*) that minimize time and/or distance (*Ghose et al., 2006*). It also helps mitigate manoeuvring problems likely to occur during pure pursuit as the inter-party distance approaches zero (particularly where pursuer speed is much greater than that of the prey) (*Siouris, 2004*).

The development of the chase for a cursorial predator operating in the planer environment of the ground, however, invariably starts with a stalking phase on essentially non-moving prey followed by the rush (*Estes and Goddard, 1967*; *Elliot et al., 1977*; *Mills, 1990*; *Williams et al., 2014*). This situation is very different from some modelled interactions where both predators and prey have constant speed (e.g., *Weihs and Webb, 1984*). Two options would seem appropriate for predator-prey encounters that start prey at, or close to, zero speed in planer environments faced with a predator moving rapidly towards them. One, which is more appropriate where the predator speed is markedly higher than prey and the distance between them small, involves prey escaping in a trajectory that is perpendicular to that of the predator (*Weihs and Webb, 1984*; *Stankowich and Coss, 2007*; *Domenici et al., 2011*). In part, this 'side step' strategy capitalizes on the speed-linked smaller turn radius of the prey compared to the predator. Otherwise, if prey react to a distant, but approaching, predator by generally moving away, the speeds of both parties will tend to become better matched over time. Here, the best strategy is for prey to continue to move directly away because any other

direction will reduce the inter-party distance correspondingly (cf. *Domenici et al., 2011*). This explains observations reporting that prey typically do indeed initially run directly away from the predators (cf. *Finney et al., 1997*; *Broom and Ruxton, 2005*; *Eilam, 2005*) although this is not always the case (see review in *Domenici et al., 2011*). This manoeuvre puts the pursuer directly behind the prey. During the ensuing pursuit phase, straight line pursuit by the predator towards the prey is the only viable predator option when the prey travels in a straight line (ignoring topographic complexity —see 'Discussion'), which is also what has been observed (*Shubkina et al., 2012* and refs therein). As the predator-prey distance tends towards zero, the prey must turn to avoid capture, and may even have particular strategies such as stotting to enhance this (*Stankowich and Coss, 2007*). We demonstrate later that there is considerable selection pressure on the prey to time this turn so that it occurs close to the contact point (but not so close that the predator catches it). This scenario reduces differences, in both trajectories and time to contact, between the different pursuit strategies (see above) although any predictive pursuit options typically have a reduced trajectory length and time to contact than pure pursuit (see below). Differences between the pursuit strategies are also likely reduced by prey reaction to the approaching predator based on the prey having properly informed assessment of predator trajectory. In this regard, typical terrestrial 'prey', such as ungulates, have large panoramic visual fields with laterally facing orbital margins (*Heesy, 2004*) and so may be able to assess the movements of their pursuers as well as their pursuers can assess them. Indeed, escape trajectories may be modulated to keep predators within the prey's visual (or other sensory) fields (*Domenici et al., 2011*). In any event, the substantive differences between terrestrial and fluid media interactions may partly account for the reason that *Shneydor (1998)* considers that many terrestrial cursorial predators adopt pure pursuit (hence the name 'hound-hare pursuit'), something that is backed up by observation (*Schaller, 2009*; *Shubkina et al., 2010*, *2012*). Importantly though, quantitative differences in trajectories and pursuit times between pure- and predictive pursuit strategies do not change qualitative patterns. Thus, the essential message of the work, which considers mass effects, does not depend on our choice of pursuit strategy.

In developing a model that highlights the effect of mass on turn performance, we make a number of critical assumptions. These are;

A. That predator-prey interactions occur in a flat, homogenous area, otherwise lacking physical structure (*Marec and Van Nhan, 1977*; *Melikyan and Bernhard, 2005*) and with no differential power requirements according to trajectory direction (*Shepard et al., 2013*).
B. That only the pursuit phase of a hunting predator is considered (*Elliot et al., 1977*).
C. That distance between predator and prey is chosen as the primary focus of the model, with prey maximizing instantaneous distance (*Weihs and Webb, 1984*).
D. That both predator and prey are solitary.
E. That both predator and prey are geometrically similar.
F. That prey operate close to maximizing energy input to locomotion during the chase although the predators are not required to do so.

We represent the pursuit of prey by plotting trajectories based on vectors representing a predator moving directly towards the prey at a speed greater than that of the prey. We then modify trajectories according to the model (see below). We define the pursuit as consisting of two fundamental stages; *stage 1*, which involves straight line travel by the predator towards the prey and, if the prey is not caught immediately, straight line travel by the prey in response to this, and *stage 2*, where the prey initiates turns to reduce the chances of being caught and the predator responds by turning during pursuit.

## Stage 1, the predator moves in a straight line

It is assumed that the predator initiates the chase by starting from a concealed position and accelerating towards a stationary or near-stationary prey some distance *s* away. The predator eventually reaches a maximum approach speed and continues to close on the prey still at, or near, its initial position. The prey does not move in response until the predator approaches within a certain distance *x*, determined by its ability to sense the predator's approach and its reaction time. In this initial rush, the predator's acceleration is a function of its speed:

$$a_{pred} = f_{pred}\left(v_{pred}\right). \tag{1}$$

For example, in the case of a cheetah, this relationship would be;

$$a_{pred} = 47.041 v_{pred}^{-0.945},$$

where this is taken from the tangential acceleration vs speed graph given by Wilson et al. (**Wilson et al., 2013a**) for a calculated cheetah power of 90 W/kg. During the chase, the predator starts with an initial constant velocity $v_{pred}(0)$ and then accelerates (magnitude $a_{pred}$) up to its maximum velocity $v_{pred}(max)$. Thereafter, when the predator's speed is constant, the distance between it and the prey continues to close at a rate of $v_{pred}(max)t$. The prey's response should nominally be to react by accelerating away from the predator according to some function of speed (cf. **Equation 1**). The prey can choose to move at any angle (θ) relative to the predator's approach so that, at the end of the predator's reaction time, $t_{react}$, the heading for the prey relative to the predator (**Figure 8**) will be given by φ where;

$$\frac{D}{\sin\theta} = \frac{s_{prey}}{\sin\phi}.$$

## Stage 2; the predator turning to follow the prey

The approach given above for stage 1 is an approximation describing the best escape strategy for the prey when the distances between predator and prey are appreciable. However, as the predator-prey distance ($D$) decreases, the prey must adopt a different trajectory or inevitably be caught. We propose that this strategy can only involve a turn (which we define as a change in heading angle by the prey), for the following reasons:

i. straight line deceleration will reduce $D$ faster,
ii. in a straight line trajectory, the prey can only maximise $D$ by maximizing its speed,
iii. turning can exploit any response delay by the predator $t_{react}$ leading to the predator continuing with an inappropriate trajectory for a period, and
iv. the predator's speed must exceed that of the prey in order for it to gain ground, so that, other things being equal, the predator will have a greater turn radius and overshoot the prey (cf. [**Wilson et al., 2013a**]).

We model the condition of the predator pursuing prey, with both parties initially travelling in the same line and both travelling at constant speed, with the predator travelling faster *at* $v_{pred}(max)$.

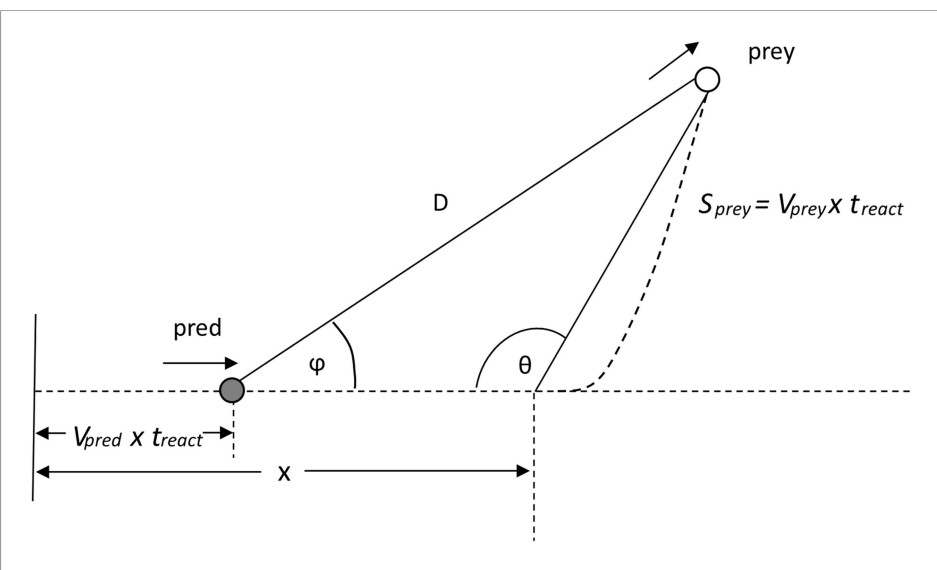

**Figure 8**. Illustration of the heading *Φ* of a prey to a predator at a time $t_{react}$ after the prey begins to move. D is the distance between the predator and prey after the predator's reaction time.

We then assume that the prey makes a turn of a given angle at this speed with the radius of the turn defined by the relationship between turn radius and speed (cf. *Equation 1*). After a defined $t_{react}$, the predator will turn in the same direction as the prey, with its own turn radius being dependent on its speed. We modelled the subsequent motion of the predator using the following assumptions:

A. The predator will initially move towards the prey using its minimum turn radius at that speed and will continue this turn until such time as its velocity vector is aimed directly at the prey.
B. Thereafter, the predator will aim directly at the current position of the prey, until such a time as the prey is either caught or the prey turns in a new direction (*Figure 9*).

We note that the predator may adopt strategies other than direct pursuit, such as aiming for the place that the prey is predicted to be after a defined time. While these strategies lead to differences in distance run by the predator until contact with the prey, or time to intercept, they do not change the general functional response in the variables we examine for the purposes of examining the effect of mass on strategies.

## Animal body mass, muscle strength and turning ability

The muscle mass and body mass of an animal is proportional to $L^3$;

$$M \propto L^3, \tag{2}$$

where $L$ represents one of the (equal) linear dimensions of the animal assuming that animals may be represented in this manner. The muscle force development (almost independent of animal type (*Schmidt-Nielsen, 1984*)) is proportional to $L^2$ (because the force is proportional to the cross sectional area of the muscle [*Schmidt-Nielsen, 1984*]) so that;

$$F_m \propto L^2 \tag{3}$$

The centripetal force required for a turn moving at speed $v$ is;

$$F_c = \frac{Mv^2}{r}, \tag{4}$$

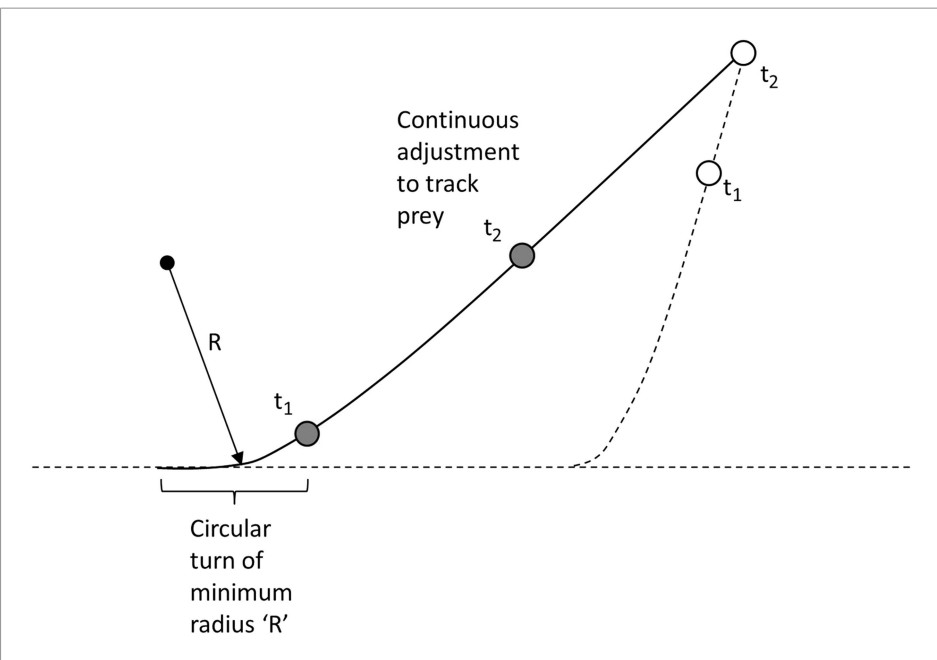

**Figure 9**. Modelling the predator's movement as an initial turn of minimum radius followed by continuous alteration of direction to track moving prey.

where $r$ is the radius of the turn.

Clearly, the ultimate turning ability of an animal is limited by the mechanical strength of its bones, muscles and tendons since this determines the maximum force available to the animal that can be applied to the ground. However, for a simplistic and approximate model, we propose here that the force available is $kL^2$ where $k$ is a constant representative of the material making up the limb. For example, the ultimate compressive strength of compact bone is $170 \times 10^6$ N/m$^2$, the tensile strength of ligament and tendon is $50$–$100 \times 10^6$ N/m$^2$, while these values for muscle are lower, in the range $10^5$–$10^6$ N/m$^2$. It follows that, when an animal is taking a turn at the minimum radius of curvature at that speed,

$$\frac{mv^2}{r} = kL^2 \text{ so that } k = \frac{\rho L v^2}{r}, \tag{5}$$

where we assume that $k$ is nominally a constant (although there will be appreciable variability given the variation in animal morphology). However, for example, taking a cheetah of mass 30 kg running at 20 m/s, with a minimum turn radius of 40 m (*Wilson et al., 2013a*), $k$ would be approximately 3100 and, for convenience, we use this value in our model (noting assumption (e) above).

## Field trials to examine model predictions on free-living cheetahs

We examined the perspectives from our models with respect to prey pursuit behaviour observed directly, and quantified using collar-fitted tri-axial accelerometers (G6a, Cefas, UK —recording rate 30 Hz), on six free-ranging cheetahs in the Kgalagadi Transfrontier Park (25°46′S 20°23′E), southern Africa. Cheetahs, which varied in mass between 30 and 45 kg, were equipped for a total of 66 animal days during which 36 pursuits of prey were observed.

The orthogonal, tri-axial acceleration data were first sorted to identify the periods of active pursuit by matching times with periods when the animals were observed to hunt. Data from the sway and heave axes were somewhat interchangeable due to a partially rotating collar, but co-varied directly

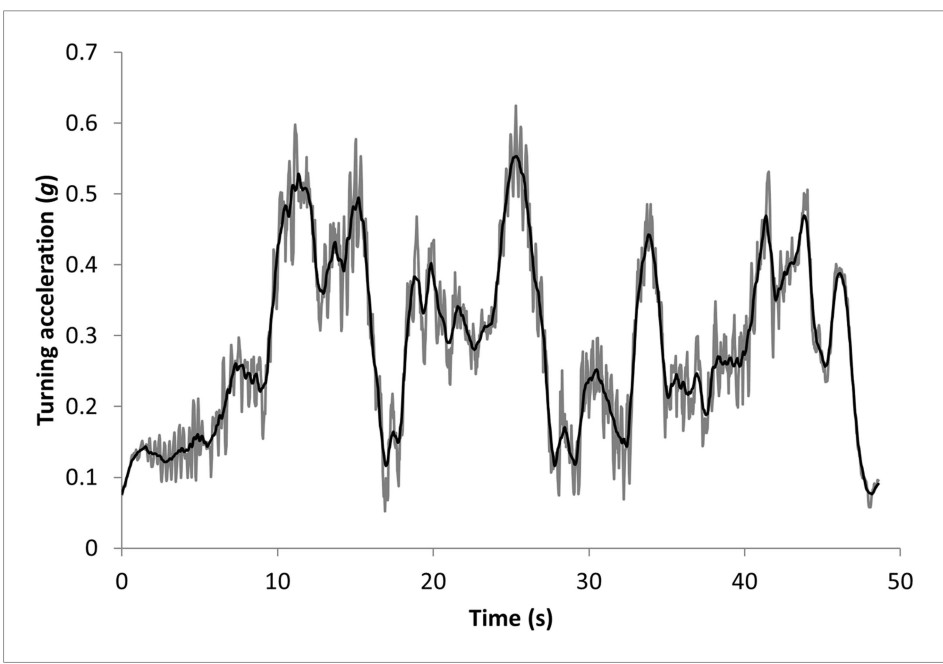

**Figure 10**. Data showing the vectorial sum of the static heave and sway acceleration axes (expressed as a departure from 1) during the pursuit phase of a cheetah hunting a steenbok. The grey lines show the values using a running mean window of 2 s, which still shows appreciable signal noise due to the bounding movement of the animal. These values have been further smoothed over 0.5 s to give the black line which shows the main features of the lateral acceleration during turns with maximum g-forces developed during turns displayed as peaks.

and changed with lateral acceleration and thus cornering behaviour (the surge axis varied with longitudinal acceleration or deceleration). Thus, the mediolateral acceleration data (heave and sway axes) were initially smoothed over 2 s to derive a measure of the static (nominally gravity-based) acceleration (*Shepard et al., 2008*) and were found to show obvious waveforms due to the bounding gait of the cheetahs (*Figure 10*). These data were thus further smoothed over 0.5 s to minimize the influence of these waveforms while not overly compromising smoothed values (*Figure 10*). Turns were identified as increasing departures of the doubly smoothed data from 1×*g*. The specific points of their maxima were identified by values that departed maximally from 1.0. Immediately adjacent (and lower) peaks which occurred as a result of the bounding behaviour (which could be readily identified in the single-smoothed data—see *Figure 10*) were precluded.

Analyses of these data were performed in R version 3.0.2 (*R Core Team, 2013*). The relationship between prey species, hunt success and turn number on turn rate was determined using a general linear model (*Bates et al., 2013*). Turn number was entered as a covariate, prey species and hunt success as factors and cheetah ID as a random factor to account for repeated measurements within animals. The relationship between turn number and hunt success on turn duration was determined using a similar model with turn number entered as a covariate, hunt success as a factor and cheetah ID a random factor. For both models, function 'lmer' was used in the package lme4. Wald $\chi^2$ statistics and p values were obtained using the function 'Anova' in the package 'car'. Data were tested for normality and homoscedasticity of variance using Shapiro–Wilk and Levene's tests.

## Trends in turn performance with respect to predator and prey size within mammals

We used the approach outlined in our model to undertake a broad-based analysis to examine how the mass of terrestrial predators and prey (using ranges between 0.05 and 5000 kg) affected the interplay of maximum speed and minimum turn radius at that speed. Predictions for maximum speed were derived from an allometric relationship between mass (M, kg) and speed (V, km/h) for mammals of (*Garland, 1983*);

$$\log_{10}V_{max} = 1.478 + 0.2589(\log_{10}M) - 0.0623(\log_{10}M)^2.$$

We then took data from an extensive data base on the mass of mammalian predators and their mammal prey (*Carbone et al., 1999*) so as to place data from wild animals in this context to determine whether any trends were apparent.

## Acknowledgements

This study was supported by the Royal Society (2009/R3 JP090604) and NERC (NE/I002030/1) to DMS. We thank SANParks and the Department of Wildlife and National Parks, Botswana for allowing our research in the KgalagadiTransfrontier Park (Permit Number 2006-05-01 MGLM) and from The Lewis Foundation, South Africa, The Howard G Buffet Foundation, National Geographic, Kanabo Conservation Link, Comanis Foundation, Panthera and the Kruger Park Marathon Club for financial support to MGLM.

## Additional information

### Funding

| Funder | Grant reference | Author |
|---|---|---|
| Royal Society | 2009/R3 JP090604 | David M Scantlebury |
| Natural Environment Research Council (NERC) | NE/I002030 | David M Scantlebury |
| Grigg-Lewis Foundation, Inc. | NA | Michael GL Mills |

The funders had no role in study design, data collection and interpretation, or the decision to submit the work for publication.

## Author contributions

RPW, IWG, Conception and design, Analysis and interpretation of data, Drafting or revising the article; MGLM, Analysis and interpretation of data, Drafting or revising the article; CC, Analysis and interpretation of data, Drafting or revising the article, Contributed unpublished essential data or reagents; JWW, Acquisition of data, Analysis and interpretation of data, Drafting or revising the article; DMS, Conception and design, Acquisition of data, Analysis and interpretation of data, Drafting or revising the article

## Ethics

Animal experimentation: Permission and ethical clearance were granted by SANParks ethical and research committees to conduct the field research, Project Number 2006-05-10 MGMI. The study was performed in accordance with the commendations in the Guide for the Care and Use of Laboratory Animals of the National Institutes of Health. All immobilizationand collaring of wild animals was conducted by a registered individual (GM), under the direction of a SANParks veterinarian.

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
