## [Decision Letter]

[Editors’ note: this article was originally rejected after discussions between the reviewers, but the authors were invited to resubmit after an appeal against the decision.]

Thank you for choosing to send your work entitled “The bigger they are, the wider they turn: Mass defines hunting options in cursorial predators”: for consideration at *eLife*. Your full submission has been evaluated by Ian Baldwin (Senior Editor), a Reviewing Editor, and three peer reviewers, and the decision was reached after discussions between the reviewers. Based on our discussions and the individual reviews below, we regret to inform you that your work will not be considered further for publication in *eLife*.

In addition to the reviews that you can find below, there was a lively discussion that followed the initial submissions of the reviews, and we wanted to capture some of the salient points of this discussion, in the hope that this may of assistance to you. First, we wanted to emphasize that there was overall enthusiasm for the simplicity and elegance of the approach, but that perhaps more details need to be added to the model. In addition to broadening the focus of the paper taxonomically, including additional complexities of the predator-prey pursuit, in particular the element of endurance would greatly broaden the appeal of the analysis.

One of the reviewers recommended as a starting point: “the classic [53], Missile Guidance and Pursuit, which deals with the overshooting problem with pure pursuit strategies described in the current manuscript explicitly. The other is Siouris 2003, Missile Guidance and Control Systems. That deals with this problem in chapter 4 where Siouris writes: ‘Pursuit guidance is considered impractical as a homing guidance law against moving targets because of the difficult maneuvers that are required to end the attack in a tail chase. That is, the maneuvers required of the missile become increasingly hard during the last, critical, stages of the flight. Another disadvantage of this guidance method is that the missile speed must be considerably greater than that of the target. The sharpest curvature of the missile flight path usually occurs at the end of the flight, so that at this time the missile must overtake the target. If the target attempts to evade, the last-minute angular acceleration requirements placed on the missile could exceed the aerodynamic capability, thereby causing a large miss distance’, which rather captures the situation in the manuscript.”

Reviewer #1:

In this manuscript the authors modelled predator-prey interactions with a focus on body-mass differences and how these differences constrain hunting and escape behaviour. A number of simplifying assumptions are made to break down the interaction into simple components and the predictions are compared to data from the literature (including recent novel data on locomotion performance obtained from sophisticated tags on moving animals).

In the model the authors consider primarily speed and cornering ability (as a function of mass). When comparisons with real data from predators and prey are made, it seems obvious that differences between predator and prey in endurance should also play an important role especially given that you considered the full duration of hunts. Endurance is part of the discussion but does not form part of the model generating predictions.

Another factor that strikes me as potentially important is the ability to anticipate turns of prey. But I suppose this is partly covered by the authors' consideration of reaction time. Nevertheless if predator can well anticipate cornering of their prey then this could compensate partly of the disadvantage of greater turning angles due to greater mass.

*Reviewer #2*:

This is an interesting theoretical paper that takes into account simple biomechanical principles in order to explain the behaviour of predators and prey. Manoeuvrability is certainly a major factor in predator-prey gambits, although it is largely overlooked in certain taxa and therefore this manuscript is potentially a welcome addition to the literature. My main comment is that this paper really mainly applies to terrestrial mammals, and the authors should clarify this point right from the title and the Abstract. A lot of work on manoeuvrability and predator-prey interactions has been done on aquatic and aerial species. Some of that work may actually be relevant for this manuscript as well, at least in the discussion for comparing various predator-prey situations.

Among other, potentially relevant work (both theoretical and experimental) on the effect of manoeuvrability (and scaling, in some of the manuscripts listed below) on predator-prey interactions of non-terrestrial animals is: van den Hout (Behav Ecol, 2010), Hedenstrom and Rosen (Behav Ecol, 2011), Coombs et al. (J exp Biol, 2011), Warrick (C J Zool, 1998), Weihs and Webb (J Theor Biol, 1984), Weihs (Fisheries Bulletin, 1981), Webb and DeBuffrenil (Tran Am Fish Soc, 1990), Domenici (Comp Biochem Physiol, 2001; J exp ; Biol, 2011), F. E Fish (Performance constraints on the manoeuvrability of flexible and rigid biological systems, pp. 394–406 in: Proceedings of the Eleventh International Symposium on Unmanned Untethered Submersible Technology).

While the citations in most of the paper are on terrestrial animals, [65] is on fish. I think the authors should either: 1) expand their view by taking into account aerial and aquatic species as well or 2) more clearly choose to deal with terrestrial animals only.

Introduction: The scaling trend with body mass has been examined in aerial and aquatic species. This should be clarified here.

Introduction: Substitute “mammals” with “terrestrial mammals”

Results: Here the authors analyse two cases: 1) the predator cuts corners and 2) the predator overshoots. What about the case of a predator that manages to follow the prey precisely? (Is this the case of a very short reaction time? This could be clarified).

Cornering should be defined.

Figure 4 and Table 1: The stats are a little bit confusing. The authors should specify what they are comparing as well as the degrees of freedom.

The authors suggest that the best situations for a predator are those when 1) the prey makes multiple rapid turns or 2) the prey runs in a straight line. This seems contradictory and further explanations (perhaps an additional figure) would help.

Discussion: The authors mention topography and vegetation as one of the factors that can alter the game rules. This is an important point and deserves a little bit more discussion.

Model: The scaling of turning radius also depends on the scaling of maximum speed. Predators or prey may however decide to run slower in order to produce a tighter turning radius. Alan Wilson et al. (2013) for example suggest that peak running speed is unlikely to be used by cheetahs in the final stage of their successful hunts because it would imply poor manoeuvrability since at this speed cheetahs would take as long as 6 s to perform a 180^0^turn.

There are further general considerations about speed and manoeuvrability that could be made in the Discussion, for example about the trade-off between speed and accuracy (12) as well as the principles of unpredictability (Jones et al. Behav ecol, 2011).

Reviewer #3:

The paper is interesting, the derivation of a simple scaling law for predator prey pursuit interactions is appealing, but the authors use an extraordinarily naïve pursuit law. This is really surprising because there is a deep and rich literature dealing with the pursuit problem, from Achilles and the tortoise to modern missile systems. That literature is really interesting and directly applicable. The only case they have analysed is a very simplified version of 'pure pursuit' where the predator aims at the current position of the prey, and their main conclusion is a very well known problem for pure pursuit – the closing stages of the trajectory require increasing turn rates from the pursuer. This has been demonstrated from geometry. Thus the main conclusion from the paper is not novel.

[Editors’ note: what now follows is the decision letter after the authors submitted for further consideration.]

Thank you for resubmitting your work entitled “Mass limits hunting options in terrestrial, cursorial predators” for further consideration at *eLife*. Your revised article has been favorably evaluated by Ian Baldwin (Senior Editor), a Reviewing Editor, and one other reviewer. The manuscript has been improved but there are some remaining issues that need to be addressed before acceptance, as outlined below:

*Reviewer #2*:

The paper is now much improved. I still have a few comments that the authors should be able to deal with.

1) Title: Now the title includes the term “terrestrial” which clarifies what type of animals are considered. However, the current title is a little bit vague, because the authors have removed the sentence “The bigger they are, the wider they turn”. I suggest reinstating it, therefore using this full title “The bigger they are, the wider they turn: Mass defines limits hunting options in terrestrial, cursorial predators”.

2) Introduction: Across these terrestrial studies. Rewrite as “across these studies on terrestrial animals”.

3) Results: “A third scenario might be where the predator follows the prey trajectory precisely, in which case there is no benefit to either party.” This is not immediately clear (probably because it takes into account manoeuvrability only). If a predator follows a prey precisely, it will eventually catch it as long as its speed is higher than that of the prey. I suggest rephrasing this sentence and adding a panel in Figure 1, that takes into account this simple possibility.

4) “Although a single turn may not lead to capture or escape of the prey, multiple turns with the same outcome will can do so (Figure 1).” It is not clear what type of outcome the authors are referring to.

5) The authors reply that “cornering” was defined in the Methods section. I could not find this definition.

6) Discussion: Work by [34] is not in fluid media. It was done using computer videos and human subjects as predators (volunteer students). The theory behind protean behaviour (see also the seminal work by Humphries and Driver in the ‘70 s) implies catching the predator by surprise, for example by turning left just after a right turn. Increasing the amount of turning may make the path of the prey more unpredictable, (e.g. by using random turns just before the critical phase, as suggested by the authors) and this is a general consideration that may apply also to a 2D environments.

7) Methods: “Thus, starting from a speed of zero, any direction taken by the prey as a response to this, that is not directly away from the predator, will reduce the inter-party distance correspondingly.” The situation is not so simple. Previous models (e.g. [70]; [16]) suggest that the optimal angle of escape for a prey depends on the ratio of predator speed/prey speed. For prey speeds higher than predator speed, the prey should escape directly away from the predator. However, for predator speeds that are higher than the prey speed, the prey should escape at 90° from the predator attack path.

8) In addition, stotting may allow a sudden change in escape direction, unlike running. This strategy is common in some ungulates (e.g. [57]).

9) Methods: “which explains why observations report that prey typically do indeed initially run directly away from the predators”. While this is generally true, it is not always the case. See for example review by [16] cited in this manuscript.

10) Methods: “In this regard, typical terrestrial 'prey', such as ungulates, have large panoramic visual fields with laterally facing orbital margins (29) and so may be able to assess the movements of their pursuers as well as their pursuers can assess them.” In some prey, escape trajectories may aim at keeping the predator just within the limits of the prey's visual field (16). Therefore, not all angles of escape may be the same, for a prey chased by a predator. There may be optimal ones, from a sensory perspective, that allow keeping track of the danger. Indeed, if a prey has a blind zone posteriorly, turning itself could be a way to keep track of where the predator is. While this paper is mainly about the biomechanical principles that may determine predator-prey strategies, some considerations on the sensory systems of predators and prey would increase the ecological and behavioural relevance of this work.

---

## [Author Response]

[Editors’ note: the author responses to the first round of peer review follow.]

*In addition to the reviews that you can find below, there was a lively discussion that followed the initial submissions of the reviews, and we wanted to capture some of the salient points of this discussion, in the hope that this may of assistance to you. First, we wanted to emphasize that there was overall enthusiasm for the simplicity and elegance of the approach, but that perhaps more details need to be added to the model. In addition to broadening the focus of the paper taxonomically, including additional complexities of the predator-prey pursuit, in particular the element of endurance would greatly broaden the appeal of the analysis*.

We have now clarified elements of the model in some detail (see below) and, following the advice of reviewer #2, limited our treatment to terrestrial predators. This allows us to keep the model simple while demonstrating its salient features. We have also added the requested element of endurance and included an extra figure for this (see below).

*One of the reviewers recommended as a starting point: “the classic*
[53]*, Missile Guidance and Pursuit, which deals with the overshooting problem with pure pursuit strategies described in the current manuscript explicitly. The other is Siouris 2003, Missile Guidance and Control Systems. That deals with this problem in chapter 4 where Siouris writes: ‘Pursuit guidance is considered impractical as a homing guidance law against moving targets because of the difficult maneuvers that are required to end the attack in a tail chase. That is, the maneuvers required of the missile become increasingly hard during the last, critical, stages of the flight. Another disadvantage of this guidance method is that the missile speed must be considerably greater than that of the target. The sharpest curvature of the missile flight path usually occurs at the end of the flight, so that at this time the missile must overtake the target. If the target attempts to evade, the last-minute angular acceleration requirements placed on the missile could exceed the aerodynamic capability, thereby causing a large miss distance’, which rather captures the situation in the manuscript*.*”*

Both the references mentioned by the reviewer are excellent (we were aware of Shneydor but not Siouris) and we have incorporated them into the manuscript now, expanding our consideration of pursuit strategies considerably in the Methods. A major mistake (pointed out by reviewer #2) was that we were not explicit about only dealing with terrestrial pursuit systems, which are fundamentally different with regard to manoeuvrability and speed to aerial systems. The issue of missile manoeuvres becoming more difficult towards the end of the chase is obviously critical for flight pursuit, not least because airborne predators (or missiles) cannot reduce speed without losing lift, with all the stability and turn radii complications that this engenders. In addition, missiles typically have much greater speed than their targets, which is not the case in many terrestrial encounters. Having now clearly specified that we only consider terrestrial predators, many of these particular problems become less critical (terrestrial predators actually have smaller turn radii with decreasing speed and are liable to have greater stability and accuracy – see e.g. [12] and references therein now cited in our manuscript).

Actually, we did consider strategies other than ‘pure pursuit’ in earlier iterations of the manuscript but we removed them in an attempt to present the most simple scenario. This was obviously a mistake. Clearly, because different pursuit strategies change trajectories, they may be expected to change the conclusions drawn regarding the primary focus of the manuscript; that of the effects of mass on turn radius. In fact, the general conclusions do not change because all terrestrial animals have operate within minimum turn radii rules which are critically and profoundly affected by mass and these define the major features of turn gambits. However (and with hindsight this seems obvious now), we should have discussed this in the manuscript because otherwise readers might wonder whether we were being naïve (as the reviewer did).

Reviewer #1:

*In the model the authors consider primarily speed and cornering ability (as a function of mass). When comparisons with real data from predators and prey are made, it seems obvious that differences between predator and prey in endurance should also play an important role especially given that you considered the full duration of hunts. Endurance is part of the discussion but does not form part of the model generating predictions*.

This is a very fair point, not least because predator-prey pursuits consist of more than one turn gambit (as our own data from cheetahs make clear). We were simply worried about ‘over-cooking’ our model. We have now incorporated the issue of endurance by providing a simple metabolic power diagram that highlights overall accumulating energy costs during a pursuit according to speed, turns and straight-line sections in pursuits. Although the metabolic cost of turning has been quantified for humans (by us – see [74] – Ecology Letters, cited in the manuscript), and found to be substantial, our study did not quantify the relationship between angular velocity and VO_2_, which would have been most useful for insertion into the model. Our solution now simply presents a single scenario of a chase consisting of multiple turns and straight-line sections at fixed speeds for both predators and prey. A new figure illustrates how over- or under-shooting by the predator may affect the outcome of the chase where endurance is represented by available energy limits.

Incidentally, we have now just repeated the ‘VO_2_ versus turning’ experiment that we published in Ecology Letters, this time varying speed and angular velocity using 22 participants. Thus, we will shortly be in a position to define the relationship between the variables more completely.

*Another factor that strikes me as potentially important is the ability to anticipate turns of prey. But I suppose this is partly covered by the authors' consideration of reaction time. Nevertheless if predator can well anticipate cornering of their prey then this could compensate partly of the disadvantage of greater turning angles due to greater mass*.

The issue of predicting prey trajectory is now included in the manuscript (see remarks to reviewer #3). Currently, we point to the fact that whether predators employ pure pursuit or some predictive pursuit rules does not change the fundamental nature of the chase consisting of a series of straight line sections and turns (although it does change the details of the turn trajectory). However, the issue of overshoot or undershoot remains the same irrespective of pursuit strategy (as reviewer #3 points out, this general phenomenon has been known from geometric studies for some time) and where it becomes specifically of interest to us is how the mass affects the degree of overshoot and/or undershoot.

Reviewer #2:

*This is an interesting theoretical paper that takes into account simple biomechanical principles in order to explain the behaviour of predators and prey. Manoeuvrability is certainly a major factor in predator-prey gambits, although it is largely overlooked in certain taxa and therefore this manuscript is potentially a welcome addition to the literature. My main comment is that this paper really mainly applies to terrestrial mammals, and the authors should clarify this point right from the title and the Abstract. A lot of work on manoeuvrability and predator-prey interactions has been done on aquatic and aerial species. Some of that work may actually be relevant for this manuscript as well, at least in the discussion for comparing various predator-prey situations*.

This is a very pertinent point and it was, in fact, only on re-reading it from the ‘non-terrestrial’ perspective, that we realised how incompletely it was framed. We have now made multiple changes throughout the manuscript (including in the title) to make it explicit that we only deal with terrestrial cursorial animals. We have also, though, incorporated many references from aerial and aquatic species in ‘setting the scene’ (see also comments below).

*Among other, potentially relevant work (both theoretical and experimental) on the effect of manoeuvrability (and scaling, in some of the manuscripts listed below) on predator-prey interactions of non-terrestrial animals is: van den Hout (Behav Ecol, 2010), Hedenstrom and Rosen (Behav Ecol, 2011), Coombs et al. (J exp Biol, 2011), Warrick (C J Zool, 1998), Weihs and Webb (J Theor Biol, 1984), Weihs (Fisheries Bulletin, 1981), Webb and DeBuffrenil (Tran Am Fish Soc, 1990), Domenici (Comp Biochem Physiol, 2001; J exp* ; *Biol, 2011), F. E Fish (Performance constraints on the manoeuvrability of flexible and rigid biological systems, pp. 394–406 in: Proceedings of the Eleventh International Symposium on Unmanned Untethered Submersible Technology)*.

These are excellent suggestions, particularly for the framework. We have now incorporated almost all of these (which necessitates an expanded literature section but one which, we feel, helps create a more solid contribution).

*While the citations in most of the paper are on terrestrial animals,*
[65]
*is on fish. I think the authors should either: 1) expand their view by taking into account aerial and aquatic species as well or 2) more clearly choose to deal with terrestrial animals only*.

Given the extensive literature on predator-prey interactions in fluid media and the substantial differences between fluid medium locomotion and terrestrial locomotion (to which we now refer), we have adopted the reviewer’s suggestion to define that we only deal with terrestrial animals. Note that we have, though, included much of suggested literature in the introduction to provide a more complete overview of the problem.

*Introduction: The scaling trend with body mass has been examined in aerial and aquatic species. This should be clarified here*.

This has now been done.

Introduction: Substitute “mammals” with “terrestrial mammals”

Done.

Results: Here the authors analyse two cases: 1) the predator cuts corners and 2) the predator overshoots. What about the case of a predator that manages to follow the prey precisely? (Is this the case of a very short reaction time? This could be clarified).

This is an important point because it appears that we have not given this option consideration. This has now been clarified.

*Cornering should be defined*.

This has now been done in the Methods section.

Figure 4
*and*
Table 1*: The stats are a little bit confusing. The authors should specify what they are comparing as well as the degrees of freedom*.

We have now clarified the stats.

*The authors suggest that the best situations for a predator are those when 1) the prey makes multiple rapid turns or 2) the prey runs in a straight line. This seems contradictory and further explanations (perhaps an additional figure) would help*.

We have now elaborated on this but have not introduced an extra figure since we already have an extra figure detailing energy use during extended chases (see reply to reviewer above).

Discussion: The authors mention topography and vegetation as one of the factors that can alter the game rules. This is an important point and deserves a little bit more discussion.

This is indeed important and has now been expanded.

*Model: The scaling of turning radius also depends on the scaling of maximum speed. Predators or prey may however decide to run slower in order to produce a tighter turning radius. Alan Wilson et al. (2013) for example suggest that peak running speed is unlikely to be used by cheetahs in the final stage of their successful hunts because it would imply poor manoeuvrability since at this speed cheetahs would take as long as 6 s to perform a 180*^*0*^
*turn*.

This point has now been incorporated into the text.

*There are further general considerations about speed and manoeuvrability that could be made in the Discussion, for example about the trade-off between speed and accuracy (*[12]*) as well as the principles of unpredictability (Jones et al. Behav ecol, 2011)*.

Both these points have now been incorporated into the text.

Reviewer #3:

*The paper is interesting, the derivation of a simple scaling law for predator prey pursuit interactions is appealing, but the authors use an extraordinarily naïve pursuit law. This is really surprising because there is a deep and rich literature dealing with the pursuit problem, from Achilles and the tortoise to modern missile systems. That literature is really interesting and directly applicable. The only case they have analysed is a very simplified version of 'pure pursuit' where the predator aims at the current position of the prey, and their main conclusion is a very well known problem for pure pursuit – the closing stages of the trajectory require increasing turn rates from the pursuer. This has been demonstrated from geometry. Thus the main conclusion from the paper is not novel*.

The referee makes a very valuable point and also makes another problem apparent to us. Many thanks for highlighting the necessity of referring to the first of these, specifically the pursuit law issue, but also, indirectly making us aware of a second issue (see below).

In fact, in an earlier version of this manuscript, we did consider various pursuit law options, documenting how they changed chase parameters. The reason that we dropped this was that, although different pursuit strategies result in different predator trajectories and different times to predator-prey contact (see Figures 11, 12 and 13), they did not change the essence of the functional response of distance between predator and prey according to whether the predator under-cut or overshot the prey trajectory (Figure 13) (as the referee points out, this is well known from geometric studies). Clearly this needs to be stated explicitly within the manuscript and we have now done this at length, referring to the literature that the referee recommended (as well as introducing others).

Author response image 1.Trajectories for a predator starting a chase at 55 m from prey (greater than the predator’s minimum radius of turn, 40 m) showing (by the continuous line) the predator turning to follow the prey with its minimum turn radius until the prey is directly ahead and then adjusts its direction continuously to keep the prey dead ahead and (by the dashed line) the predator turning with its minimum turn radius of 40 m until it is in a direction that corresponds to the anticipated point of interception with the prey whereupon it continues in a straight line (constant bearing) to that point. Predator and prey speeds are 20 and 15 m/s, respectively. Times until interception are shown within the plot***.*****DOI:**
http://dx.doi.org/10.7554/eLife.06487.016

Author response image 2.Conditions as in Figure 11, except that the predator starts at 30 m from the prey, leading to an overshoot (note that the anticipatory chase has been shifted 20 m in the x-direction for clarity)**DOI:**
http://dx.doi.org/10.7554/eLife.06487.017

Author response image 3.Time vs distance between predator and prey plot for the overshoot scenario depicted in Figure 12 for direct (blue dashed line) and anticipatory (red continuous line) pursuits. Note how, in all scenarios, although the two different pursuit strategies result in different values of the outputs, they do not change the fundamentals in the form of either the trajectories or the distance vs time plots (cf. Figure 2 in the manuscript) with either over-shoots or corner-cutting depending on turn radius.**DOI:**
http://dx.doi.org/10.7554/eLife.06487.018

Importantly though, given that the degree of this under- or overshoot is profoundly affected by mass (the main purpose of our investigation), we concluded that it was easiest to present just one pursuit scenario rather than clutter up the work with multiple plots and outputs that showed no great differences between them within the context of the issues being discussed (note that Figure 1 in the manuscript is essentially schematic and Figures 3, 4, 5 and 6 do not contain pursuit strategy-dependent outputs). We ended up choosing the pursuit rule that seemed most supported by the literature for terrestrial running animals, direct pursuit. However, the referee does make a vital point, (**A**) that readers need to know that we have considered various options and (**B**) that choice of option does not change the overall patterns that we use to progress to the nub of the matter, namely that of predator and prey mass.

The second issue (see above) is that we clearly did not emphasize enough that the main purpose of the work was not to identify, reiterate or copy the geometric conclusions shown by others (although we are obliged to consider them to set the groundwork, hence the election of our model), but to examine the specific effect of mass on the possible strategies and outcomes in gambits between running predators and prey. We have now modified the text to try and make our primary aim clear.

[Editors’ note: the author responses to the re-review follow.]

Reviewer #2:

*The paper is now much improved. I still have a few comments that the authors should be able to deal with*.

*1) Title: Now the title includes the term “terrestrial” which clarifies what type of animals are considered. However, the current title is a little bit vague, because the authors have removed the sentence “The bigger they are, the wider they turn”. I suggest reinstating it, therefore using this full title “The bigger they are, the wider they turn: Mass defines limits hunting options in terrestrial, cursorial predators”*.

We had included “The bigger they are, the wider they turn” in the title but we were advised to avoid a two-part title.

2) Introduction: Across these terrestrial studies. Rewrite as “across these studies on terrestrial animals”.

This has now been changed.

*3) Results: “A third scenario might be where the predator follows the prey trajectory precisely, in which case there is no benefit to either party.” This is not immediately clear (probably because it takes into account maneuverability only). If a predator follows a prey precisely, it will eventually catch it as long as its speed is higher than that of the prey. I suggest rephrasing this sentence and adding a panel in*
Figure 1*, that takes into account this simple possibility*.

This third possibility has been added to the paragraph and a panel added to Figure 1 that shows this course of events.

*4) “Although a single turn may not lead to capture or escape of the prey, multiple turns with the same outcome will can do so (*Figure 1*).” It is not clear what type of outcome the authors are referring to*.

This has now been corrected (and clarified).

5) The authors reply that “cornering” was defined in the Methods section. I could not find this definition.

The identification of corners was in the second paragraph in the section entitled ‘Field trials to examine model predictions on free-living cheetahs’. This section has also now been elaborated somewhat since it was indeed rather cursory.

*6) Discussion: Work by*
[34]
*is not in fluid media. It was done using computer videos and human subjects as predators (volunteer students). The theory behind protean behaviour (see also the seminal work by Humphries and Driver in the ‘70 s) implies catching the predator by surprise, for example by turning left just after a right turn. Increasing the amount of turning may make the path of the prey more unpredictable, (e.g. by using random turns just before the critical phase, as suggested by the authors) and this is a general consideration that may apply also to a 2D environments.*

Sorry, our mistake regarding [34]. We have also highlighted the potential value of protean behaviour and referenced Humphries and Driver here too.

*7) Methods: “Thus, starting from a speed of zero, any direction taken by the prey as a response to this, that is not directly away from the predator, will reduce the inter-party distance correspondingly.” The situation is not so simple. Previous models (e.g.*
[70]*;*
[16]*) suggest that the optimal angle of escape for a prey depends on the ratio of predator speed/prey speed. For prey speeds higher than predator speed, the prey should escape directly away from the predator. However, for predator speeds that are higher than the prey speed, the prey should escape at 90 degrees from the predator attack path*.

Yes, indeed, a very good point. This is particularly true when distances between predators and prey are minimal at the onset of the chase in conditions approaching the ‘sneak attack’ of e.g. Pumas (see recent publication by Terrie Williams in Science (2014)). We have modified the manuscript to incorporate this point and cited appropriately.

*8) In addition, stotting may allow a sudden change in escape direction, unlike running. This strategy is common in some ungulates (e.g.*
[57]*).*

This has now been noted in the manuscript, together with the reference.

*9) Methods: “which explains why observations report that prey typically do indeed initially run directly away from the predators”. While this is generally true, it is not always the case. See for example review by*
[16]
*cited in this manuscript*.

This has now been added, as has the citation.

*10) Methods: “In this regard, typical terrestrial 'prey', such as ungulates, have large panoramic visual fields with laterally facing orbital margins (*[29]*) and so may be able to assess the movements of their pursuers as well as their pursuers can assess them.” In some prey, escape trajectories may aim at keeping the predator just within the limits of the prey's visual field (*[16]*). Therefore, not all angles of escape may be the same, for a prey chased by a predator. There may be optimal ones, from a sensory perspective, that allow keeping track of the danger. Indeed, if a prey has a blind zone posteriorly, turning itself could be a way to keep track of where the predator is. While this paper is mainly about the biomechanical principles that may determine predator-prey strategies, some considerations on the sensory systems of predators and prey would increase the ecological and behavioural relevance of this work*.

Again, this is a very germane point, which we have now included, together with the reference to the work by Domenici et al. We have, however, kept this brief and to the point since it was leading more towards the ideal escape trajectories of the prey than to the pursuit strategy of the predator, which was the prime function of this part of the paragraph. We could expand it somewhat in the Discussion if deemed necessary.